# MMR Vaccine Attitude and Uptake Research in the United Kingdom: A Critical Review

**DOI:** 10.3390/vaccines9040402

**Published:** 2021-04-19

**Authors:** Louis Torracinta, Rachel Tanner, Samantha Vanderslott

**Affiliations:** 1Institute of Human Sciences, University of Oxford, Oxford OX2 6QS, UK; louis.torracinta1@student.lshtm.ac.uk (L.T.); rachel.tanner@ndm.ox.ac.uk (R.T.); 2The Jenner Institute, University of Oxford, Oxford OX3 7DQ, UK; 3Oxford Vaccine Group, University of Oxford, Oxford OX3 7LE, UK

**Keywords:** MMR, vaccine hesitancy, critical review, Wakefield, child immunisation, United Kingdom

## Abstract

This review critically assesses the body of research about Measles-Mumps-and-Rubella (MMR) vaccine attitudes and uptake in the United Kingdom (UK) over the past 10 years. We searched PubMed and Scopus, with terms aimed at capturing relevant literature on attitudes about, and uptake of, the MMR vaccine. Two researchers screened for abstract eligibility and after de-duplication 934 studies were selected. After screening, 40 references were included for full-text review and thematic synthesis by three researchers. We were interested in the methodologies employed and grouped findings by whether studies concerned: (1) Uptake and Demographics; (2) Beliefs and Attitudes; (3) Healthcare Worker Focus; (4) Experimental and Psychometric Intervention; and (5) Mixed Methods. We identified group and individual level determinants for attitudes, operating directly and indirectly, which influence vaccine uptake. We found that access issues, often ignored within the public “anti-vax” debate, remain highly pertinent. Finally, a consistent theme was the effect of misinformation or lack of knowledge and trust in healthcare, often stemming from the Wakefield controversy. Future immunisation campaigns for children, including for COVID-19, should consider both access and attitudinal aspects of vaccination, and incorporate a range of methodologies to assess progress, taking into account socio-economic variables and the needs of disadvantaged groups.

## 1. Introduction

Infectious disease continues to be highly relevant to public health; 2020 and the years following will undoubtedly be remembered for the historic human and economic costs of the novel coronavirus, SARS-CoV-2, which the World Health Organisation (WHO) declared a global pandemic in early March of 2020 [1]. The impact of COVID-19 (the respiratory disease caused by SARS-CoV-2) on human life and livelihoods will be substantial until herd immunity is reached which will likely require a robust vaccination campaign extending to children [2]. Childhood vaccines are a key defence strategy against many pathogens, and immunisation programs have been responsible for the eradication of smallpox and near-eradication of polio [3]. However, vaccines can be victims of their own success; high vaccination coverage has made many of the deadliest infectious diseases relatively rare in high and middle-income countries, and has led to a public perception that the severity of infectious diseases and human susceptibility to them has decreased [4]. Concurrently, parents have begun to question the need for vaccines, viewing risks associated with vaccines as being higher than those associated with the diseases they protect against; a focus on perceived safety concerns that has led to lower vaccine uptake [5].

Although a child’s vaccination status is partially explained by parental attitudes, this relationship is not as simple as one might expect, particularly because the relative importance of vaccine access and attitudes may vary between children in different contexts. In Europe in 2018, uptake of the Measles-Mumps-and-Rubella (MMR) vaccine increased, with more children vaccinated than ever before, but in the same year, a record number of individuals contracted the measles virus across the continent [6].

This paradox is likely to have occurred because progress in uptake is uneven across and within countries, and strategies that may be broadly successful in raising or maintaining uptake may not reach specific pockets of under-vaccination, resulting in localised outbreaks. In many contexts, the likelihood of vaccination uptake is determined foremost by access: where there is awareness of vaccine availability, no supply limitations, and vaccines are convenient to obtain [7]. On one side, vaccine access is the product of many economic and political variables, the quality of healthcare systems, and their ability to reach every corner of society. Therefore, contextual and socio-cultural information, which varies both within and between countries, is critical in explaining local patterns of vaccination uptake. Efforts to explain the determinants of vaccination uptake in a universal fashion are likely to fail; a more successful strategy requires attention to variation to achieve a more nuanced understanding.

On the other side, vaccine attitudes tend are considered to exist on a scale ranging from whether individuals support, accept, are hesitant, resist, reject, or oppose vaccination [7]. The forming of attitudes involves group-level influences ranging from media consumption to religious values and social norms; organisational determinants relating to accessibility, trust, and quality of the vaccination services; as well as individual determinants such as the parent’s knowledge and beliefs [3]. In Greece, socioeconomic factors such as the number of siblings and the father’s education level were the most important predictive factors for having missing or no vaccinations, whereas parental beliefs showed little predictive effect [8]. By contrast, a study in Nigeria found that partial-vaccination was most influenced by a lack of knowledge, particularly among mothers, but in completely unvaccinated children, parental disapproval of vaccines played the largest role [9].

Approval for vaccines is underpinned by trust in those promoting them, which when undermined, can lead to a re-interpretation of vaccine-related information. However, even the ways in which these re-interpretations occur are not uniform across countries; vaccine appraisal is localised and is greatly influenced by historical and socio-cultural differences. For example, the claim relating the MMR vaccine to autism was a major phenomenon in the United Kingdom (UK) due to high media coverage there, whereas claims that the Hepatitis B (HepB) vaccine was associated with multiple sclerosis was largely a French media phenomenon [10].

To better understand the determinants of a child’s vaccination status, the nature and role of parental attitudes towards vaccination, and what can be done to increase uptake, the context in which these questions are being asked should be understood. That said, conclusions made within a specific context, even if not universally applicable, can still add to a broader understanding of vaccine uptake, and prove useful in addressing the specific needs of a particular place. Given these considerations, this critical review will focus on MMR uptake and attitude research in the UK published since 2010 until 2021. The pertinent questions are: What determines parental attitudes to vaccines, and what role does this play in their child’s vaccination status? What factors lead to a decrease in uptake and an increase in outbreaks? How can these issues be rectified? These are fundamental public health questions, but their answers are complex. To indicate the rationale for these choices, it is worth briefly reviewing the scientific and historical context of MMR vaccination in this country.

## 2. Background

Before the 1961 introduction of an initial measles-only vaccine, the disease was extremely prevalent in the UK, with a peak of 693,803 cases in England and Wales in 1955 (note that notification data for Scotland is only accessible after 1968 and after 1974 for Northern Ireland) [11]. By the 1970s, uptake of a single measles-only vaccine increased but was still inadequate by public health standards (with fewer than 60% of children being vaccinated before age 2). To rectify this, national routine vaccination programs were introduced in the late 1970s [12], and by the mid-1980s, uptake increased to 80% and annual cases of measles declined below 100,000 in England and Wales [13], below 30,000 in Scotland [14], and below 2000 in Northern Ireland [15]. In 1988, the single measles vaccine was replaced by a combined measles-mumps-rubella (MMR) vaccine. Eventually a secondary “booster” dose at 4 years of age became clinically advised, which raised protection from 95% to 99.7% [16]. In 1993, British physician Andrew Wakefield and colleagues began publishing papers that suggested a since-discredited link between the MMR vaccine and diseases such as Crohn’s disease [17] and, most infamously, autism spectrum disorder [18].

The 1998 paper published in *The Lancet* about the MMR vaccine and autism spectrum disorder was retracted but the damage had already been done [18]. Numerous subsequent epidemiological studies with large sample sizes have found no evidence for a causative link between the MMR vaccine and autism [19]. Nonetheless, widespread sensationalist British media reports of the initial Wakefield studies led to a decline of public confidence in the combined vaccine [20]. Following the controversy, initial dose uptake fell from 95% in 1995 to ~80% in 2003 [21]. Herd immunity was therefore compromised, and measles cases began to rise sharply from 2007. However, uptake improved once more in the mid-2000s (with occasional outbreaks), following several successive ‘catch-up’ campaigns in which individuals who were not vaccinated on time were invited to get vaccinated, along with a gradual restoration of public trust [16]. By 2014, the UK ended endemic transmission of measles, and in 2016 the WHO officially declared that measles was eliminated, as first-dosage vaccine uptake had recovered and passed the 95% herd immunity threshold for the first time [6]. However, MMR uptake has seen some disturbances in the mid-2010s, and the UK lost its elimination status following a large 2018 outbreak of 913 confirmed cases of measles, which was associated with other outbreaks across Europe [6].

We will focus on MMR uptake and attitude research in the United Kingdom published since 2010. Focusing on MMR has several advantages: first, the UK has variable uptake of the vaccine across populations that is measurable through reliable public health data. This has produced a sizable and heterogeneous set of studies focused within one country context but one that can still produce a cohesive narrative. Second, given the varying methodologies and disciplines utilised across literatures, such a review will have the advantage of gathering potentially disparate studies that can contribute to a comprehensive understanding of the determinants of MMR attitudes and uptake.

We will aim tease out trends within this literature and synthesise key findings across different methodologies, in order to answer three primary questions:(1)What are the primary determinants of the vaccination status of a child?(2)What factors influence parental attitudes, and how do these attitudes affect their child’s vaccination status?(3)How can low uptake be rectified to avoid further outbreaks?

As we enter the 2020s there is a renewed need for vaccination to control global outbreaks in the context of the COVID-19 pandemic. Our objective is to produce a timely review of what we know about Britain’s experience of MMR, and a multi-disciplinary understanding of the factors influencing a child’s vaccination status.

## 3. Materials and Methods

### 3.1. Search Methods

To produce a critical review of studies of MMR vaccine uptake and/or attitudes towards the MMR vaccine was conducted within and pertaining to the UK, we reviewed data over a ten-year period starting from 2010. The reason for choosing this start date was because by 2010 MMR uptake was beginning to recover after an all-time low in 2003, but also the time lag between uptake and cases meant that early 2012 still saw the largest outbreak of measles since 1988 (also resulting in an increase in MMR uptake rates). Therefore, this time period concentrates on raising uptake and avoids the earlier literature focusing on uncertainty about the Wakefield claims in the 1990s and subsequent fall-out in the early 2000s, although much the literature referenced these earlier events. Data were reviewed systematically, using a multistep process, drawing on PRISMA guidelines [22]. Studies were deemed eligible if they fulfilled the following criteria:Published from 1 January 2010 until 19 February 2021—the date of the search query;Published in peer-reviewed journals only;Published in English;Consisting wholly or partially of an original survey, qualitative interview, trial, or data-analysis focusing on attitudes and uptake of the MMR vaccine (as well as closely connected subjects of access, decision making, and beliefs);The study in question took place in, or makes reference to, populations within the UK.

There are some methodological limitations to this review. The first relates to the nature of the generalised search: by design, it only included references that contained the terms ‘MMR’ or ‘United Kingdom’ (and associated synonyms) in the title or the abstract. Therefore articles that fulfilled other eligibility criteria but did not include these terms could have been missed; although, articles that do not mention MMR all in their abstract most likely do not focus on the MMR vaccine in a way that is important to this review and are unlikely to have altered its conclusions. Moreover, many other general reviews about vaccine hesitancy exist, such as Forster et al. [23], which focuses on qualitative research on all vaccines within the UK, or Tabacchi et al. [24] and Wilder-Smith and Qureshi [25], which both focus on MMR but throughout Europe. Undoubtedly, attitude or uptake surveys non-specific to MMR or the UK that are not mentioned here will have seen consideration elsewhere.

A second limitation is the timeframe and geographical constraints of the search query; there is an abundance of high-quality research that was published prior to 2010 or was not conducted in the UK but remains relevant to current understandings of MMR attitudes and uptake. This earlier work, while not included, provides a foundation for this decade’s research, and so still influences this synthesis. Therefore, we decided that the literature after 2010 was better focused on questions of uptake and attitudes, as by that time there had been a refutation of the Wakefield claims and MMR coverage had begun to recover. With regards to research conducted outside the UK, it is worth noting that Tabacchi et al. [24] conduct a systematic review of MMR determinants across Europe. They concluded that there was significant variation in the role of different determinants in MMR hesitancy, suggesting that conclusions made in one region may not be explanatory in another. Focusing on research conducted only within the UK, therefore, offers the benefit of avoiding these potentially confounding regional variations.

Search queries were made using PubMed and Scopus, with terms aimed at capturing all relevant literature focusing on attitudes and uptake about the MMR vaccine in the UK (see Appendix A for a complete list). Additionally, a strategy of ‘citation-chasing’ was employed where the reference lists of included or pertinent articles were searched for possible references that could have been missed in the databases. For example, Forster et al. [23] conducted a qualitative systematic review of UK vaccine decision-making research, but not specific to MMR. Within its references, an interview study by Johnson and Capdevila [26] was listed that had not been flagged by the database queries because neither the abstract nor the title specifically mentioned MMR. However, this study fulfilled the eligibility criteria and is therefore included. Two other references have been included using this strategy: a study relating to Gypsy, Roma, and Traveller (GRT) communities referenced in a government action plan for MMR strategy [27,28]; and an uptake trial specific to MMR and conducted in London [29] that was referenced in a clinical review of measles [16].

### 3.2. Inclusion Criteria

The search results were imported into Mendeley, and duplicates between PubMed and Scopus were removed, leaving 931 references. Three references that did not show up in the queries (but that were mentioned elsewhere in the literature) were added separately, yielding a total of 934 references. Studies were parsed manually in two rounds. In the first, abstracts and titles were scanned for relevance to the eligibility criteria and 861 records were removed, for two main reasons:(1)they had no relevance to the MMR vaccine—the abbreviation ‘MMR’ is also commonly used for ‘maternal mortality rate’ and ‘mismatch repair (proteins)’.(2)the research in question had not been conducted in the UK or was not related to attitudes or uptake (relating solely to functional and immunological mechanisms of the MMR vaccine, for example).

In all ambiguous cases, the reference in question was carried on to the next round for closer inspection, yielding 73 references that were followed-up in order to parse articles in more detail. In round two, we sought to confirm that the references in question fulfilled the selection criteria, especially clause IV: which consisted wholly or partially of an original survey, interview, trial, or data-analysis focusing on attitudes and uptake about the MMR vaccine. Following this, 33 further references were removed, predominantly because they were review articles (presenting no original research or only summarising the work of others) or pertained only to epidemiological documentation of specific measles or mumps outbreaks, which is outside the scope of this review. The data extraction was checked by another author, with no masking used. Any disagreements were resolved through discussion through regular meetings during the analysis stage. We adopted the Atkins’ approach for appraising qualitative research, which involved applying the CASP (Critical Appraisal Skills Programme) criteria [30].

The final 40 references are included in this critical review. A thematic synthesis of the included articles was undertaken, leading to the articles being grouped by methodology and theme. Two researchers analysed the data using a grounded theory approach, identifying themes as they emerged to produce an explanatory framework. This was an iterative process, to generate themes that emerged from the data. Early on we recognised a main organising factor in the literature was the methodological approach taken. We first started with the largest group of quantitatively-focus uptake studies and then followed the grouping from there. We decided upon this approach as previous literature reviews of MMR aimed to answer questions through pre-determined themes. We were more concerned with allowing the data to determine the results, following on from our starting point of two areas of focus ‘uptake and attitudes’ which involved a different methodological treatment.

## 4. Results

### 4.1. Themes 

After conducting our review (Figure 1), we identified five themes that are divided by the methodology and focus of the studies as follows: (1) Uptake and Demographics; (2) Beliefs and Attitudes; (3) Healthcare Worker Focus; (4) Experimental and Psychometric Intervention; and (5) Mixed Methods.

#### 4.1.1. Uptake and Demographics—9 Studies

The first category consisted of largely quantitative studies: nine surveys or analyses focusing only on uptake of the MMR vaccine, and potential inequalities or predictive demographic factors therein. Some of the studies used private longitudinal health data such as the Millennium Cohort Study of ~19,000 children born in 2000–1 [31,32] or the Secure Anonymised Information Linkage (SAIL) databank of 800,000 children living in Wales [33]. Others used public National Health Service (NHS) health data through the Child Health Information Systems [34,35,36], the Scottish Immunisation and Recall System (SIRS) [37], primary care data of the Clinical Practice Research Datalink (CPRD) [38] or the Cover of Vaccination Evaluated Rapidly (COVER) dataset produced by Public Health England (PHE) [39]. All of the studies used these datasets to examine a particular cross-section of children and the degree to which MMR uptake covaried with other quantitative variables. For example, Sandford et al. [39], Haider et al. [37], Hungerford et al. [34], and Baker et al. [35] focused specifically on socioeconomic factors and income inequalities, with the conclusion that areas of high deprivation (including those with low household income or areas of unemployment) were associated with lower vaccination uptake and timeliness. Hungerford et al. [34] further concluded that targeted catch-up campaigns in deprived areas could reduce the risk of outbreaks in the future.

The remaining four studies in this category were more heterogeneous. Hutchings et al. [33] examined the potential association between residential mobility and vaccine uptake, finding that there was no significant difference between children with varying levels of residential mobility. Perry et al. [36] found that children of asylum seekers in Wales had lower rates of uptake compared to the general population, albeit with some variation between different areas of Wales due to limitations in the outreach resources of each local health authority. Emerson et al. [31] focused on children with intellectual disabilities and found that they were at increased risk for low uptake, although this may be partially explained by the inverse association between intellectual disabilities and family socioeconomic position. Osam et al. [38] used primary care data from 400,000 mother–baby pairs and found that maternal mental illness, even when adjusting for deprivation factors, had a significant negative effect on routine MMR vaccine uptake.

Pearce et al. [32] sought to understand the demographics within which a 2013 government-sanctioned catch-up campaign was successful. Importantly, they concluded that minority and low-income groups were more likely to respond to the catch-up campaign because their under-vaccination status likely stemmed from access issues, whereas unvaccinated affluent families were more likely to have consciously rejected the MMR vaccine. Overall, this category of study supports the idea that group level or demographic factors such as education and income have strong associations with MMR vaccine uptake; this is a key finding that is further discussed in Section 5.3. These studies also suggest factors such as maternal mental illness or immigration status may be underexplored influences on uptake.

#### 4.1.2. Beliefs and Attitudes—14 Studies

The second and largest group of eligible literature comprises 12 studies relating to beliefs and attitudes, typically more qualitatively-focused, using smaller sample sizes and person-to-person interviews or qualitative surveys to contextualise the path to vaccination on an individual level. The majority utilised focus groups with parents or other individuals, particularly from ‘hard to reach communities’, to determine their beliefs and attitudes around MMR and the reasoning behind those attitudes.

Six papers focused on specific minority groups that were predicted to be at higher risk for low uptake, with the goal of probing potential barriers to vaccine access. Smith [40] and Newton [41] both reference a focus group conducted with 16 women from Gypsy, Roma, and Traveller (GRT) communities, finding that their understanding and attitude towards MMR vaccination is not different from that of the general population, and that disproportionate under-vaccination in this group stems chiefly from lack of flexibility in access to the NHS and poor service provision.

Similarly, Bell et al. [42] interviewed 30 Polish and Romanian minority community members and five healthcare workers (HCWs) serving them, reporting that challenges such as language comprehension and trust in the healthcare system were the drivers of low uptake, rather than mistrust in the vaccines themselves. Bell et al. [43] also conducted a similar focus group with nine Roma community members from Birmingham, Leeds, and Liverpool with much the same conclusion. Ellis et al. [44] conducted a further focus group and several individual interviews with nine GRT mothers in London, emphasising that GRT women have a strong sense of bodily autonomy and health knowledge with a dependence on generational beliefs on vaccination, and that this is occasionally at odds with the views or recommended timelines of the healthcare system they interact with.

Tomlinson et al. [45] studied the health beliefs of Somali mothers of pre-school aged children in the UK, and found that while general attitudes towards vaccines were positive, suspicion of MMR vaccination in particular was high; attitudes were strongly mediated by religious beliefs, the mothers’ personal experiences of the vaccination schedule, and their perceptions of their children’s ability to cope with vaccination.

Seven of the studies used broader population samples: Johnson et al. [26] and Tickner et al. [46] surveyed parents of pre-school aged children, finding that general anxieties about the MMR vaccine played a role in the final decision to vaccinate, and concluded that time constraints, uncertainties about the vaccine, and low engagement from general practitioners (GPs) or clear medical information mediated their choices. In surveys using generic UK parent samples [21,47,48], it was found that decisions were driven chiefly by the information consumed, understanding of the MMR vaccine as ‘safe’, and trust in medical advice; Hill [21] specifically found that practice nurses could play a role in changing attitudes when they are seen as credible sources of information. A survey of adolescents found that they had little practical understanding of MMR (because the diseases for which it confers protection are rare as a result of vaccinations) but understood that vaccines play a role in reducing the prevalence of infectious disease generally [49]. Kennedy et al. [50] studied a variety of Scottish individuals including adults, adolescents, and healthcare workers, and found that despite high uptake in these groups, uncertainties (although sometimes just minor doubts) about vaccines remained widespread, and that misinformation about the MMR vaccine—following the Wakefield controversy in particular—had also aroused fear of new vaccines, such as the human papillomavirus (HPV) vaccine.

A final study involved a qualitative survey of parents with then-unvaccinated children who had contracted clinically confirmed measles during a 2012–2013 outbreak in Merseyside [51], confirming that the Wakefield controversy drove many of the decisions not to vaccinate prior to this outbreak. However, the experience of contracting measles changed attitudes, and many parents reconsidered the relative cost/benefits of vaccination. As a whole, these qualitative studies provide important context to the decisions and ability of parents to have their children vaccinated. While often working with smaller samples, they nonetheless are a foundation to more clearly understand the specific scenarios that lead to children not being vaccinated, despite societal expectations. In particular, many highlight previously under-acknowledged access boundaries that deserve greater attention.

#### 4.1.3. Healthcare Worker Focus—4 Studies

Alongside the aforementioned study by Kennedy et al. [50] which included HCWs as a portion of the survey sample, four papers focused exclusively on the attitudes of HCWs themselves towards their role in MMR vaccination.

Redsell et al. [52] interviewed 22 health visitors, who are the most direct sources from whom parents gather official vaccination information in the UK. Health visitors have a specialised healthcare role directed at community health in the UK and often visit parents and their children at home [53]. They are nurses or midwives who concentrate on the health of pre-school age children, including ensuring they are vaccinated. Health visitors expressed difficulties in speaking to parents about vaccinations and a loss of professional confidence, especially when the child may be only a few weeks old. Many health visitors worried that parents felt they were a ‘nuisance’. There was also confusion about the particular role of health visitors in vaccination discussions (their primary role is in aiding parents with the care of new-born infants through domestic visits) as opposed to that of nurses or the child’s GP. Similarly, Hill et al. [54] interviewed 15 practice nurses about their role in promoting the MMR vaccine and what they perceived to be the most influential strategies in achieving this. The nurses reported the need to engage effectively with parents, respond to concerns, and build a rapport while also expelling myths about the vaccine. This effort requires the need for strong recall of the most contemporary vaccination evidence, such that nurses can assist parents in making the most informed decision possible.

Other studies focused on the role of HCWs and their interaction with minority communities. Bell et al. [55] interviewed 33 HCWs involved in vaccination delivery and outbreak management in three cities with large GRT communities that had experienced significant outbreaks. These HCWs reported that improving MMR uptake and properly managing outbreaks at a local level required links with, and an understanding of, underserved communities, as well as strong coordination and robust funding.

Finally, Mytton et al. [56] surveyed 998 HCWs about their attitudes and knowledge of several vaccines, including MMR. While not collecting testimonials, Mytton instead gathered ratings of confidence and knowledge about vaccines on a numerical scale. It was found that HCWs generally treat the MMR vaccine as more important than annual influenza vaccinations, which are offered to all HCWs by the NHS, although the author noted this trend may stem from MMR vaccinations only requiring one administration (with boosters). Research regarding HCWs is crucial given the role that such workers play in encouraging and administering vaccinations, and these four studies suggest that there is room for improvement, especially in improving HCW’s knowledge base about both the vaccine itself and their understanding of local underserved communities; this is something we will revisit in the discussion.

#### 4.1.4. Experimental and Psychometric Intervention—5 Studies

The search query yielded five experimental studies, with three pertaining specifically to new tools developed to support informed decision making about the MMR vaccine. Jackson et al. [57], having devised a web-based information aid adapted from an earlier Australian version developed by Wallace et al. [58], evaluated this web-based decision aid in a preliminary UK feasibility trial, and concluded that it was generally successful in increasing knowledge and reducing conflicted decision-making about MMR. A separate study by Jackson et al. [59] compared a paper MMR leaflet alone with a leaflet plus an in-person decision-support intervention, which was found to help parents act on their decision by reducing conflicts about the vaccine—significantly more parents who received live interventions reported vaccinating their child. Finally, based on the preceding two studies, Shourie and Jackson [60] conducted a follow-up randomised cluster trial of the online decision aid compared to an information leaflet, finding that the online aid was more successful in changing attitudes, and in prompting parents to act upon their new knowledge by vaccinating their child.

Altinoluk-Davis et al. [61] sought to compare the effectiveness of catch-up campaigns conducted either by school nurses within the school setting or via sign-posting to general practice (the current standard practice), and determined strong evidence for the improved efficacy of in-school campaigns, which can also reduce inequalities in MMR vaccination between children of varying socioeconomic backgrounds. The final study was chiefly focused on raising MMR vaccine uptake within an ethnically mixed and socioeconomically deprived community in urban East London [29]. It concluded that herd immunity was achievable with strategies such as care packages and financial support, a focus on higher quality healthcare, greater research into the demographics of under-vaccinated groups within the community, and the utilisation of follow-up processes towards parents whose children’s vaccinations were not up-to-date. Taken together, these studies demonstrate a range of effective tools that can be readily implemented in future efforts to raise uptake. In particular, they highlight ways in which parental hesitancy can be reduced and knowledge can be translated into action.

#### 4.1.5. Mixed Methods Studies—8 Studies

The eight remaining studies comprise a combination of the above, pairing quantitative uptake data with a qualitative understanding of individual knowledge, sociodemographic variables, or decision making. Two were predictive in nature, seeking to test tools that predict low uptake, based primarily on attitude factors. Tickner [62] developed the Immunisation Beliefs and Intentions Measure (IBIM), a questionnaire based on qualitative interviews and the theory of planned behaviour that is strongly predictive of the final intention to vaccinate. Brown et al. [63] devised a similar but more detailed attitude measurement instrument that includes sociodemographic questions, found to be psychometrically robust (it had internal consistency) and also reliably predicted vaccination decisions.

Walsh et al. [64] and Anderberg et al. [65] sought to draw out the complex relationship between individual demographic variables such as education, income level, and media consumption with final uptake outcomes. Walsh et al. [64] used information from the Child Health System database to identify and contact parents of children who reached their second birthday between July and September 2001, in a low MMR-uptake area South Wales, for a questionnaire concerning their media consumption, with particular reference to the Wakefield controversy throughout 2001. In the area of South Wales surveyed, consumption of English-language media and internet usage had a strong, negative influence on eventual perceptions and uptake of MMR, possibly playing a role in an eventual measles outbreak in the area a decade later. Anderberg et al. [65] utilised panel data across several health authorities to determine if variation in MMR vaccine uptake following the Wakefield controversy was correlated with other variables, such as education past the age of 18. These results go some way to explain aspects of vaccine denial among those who are more highly educated. Strikingly, the study found that uptake declined faster in areas of higher education, with spill-over effects to other vaccines, contrary to the more familiar pattern in which low educational achievement and socioeconomic deprivation correlate with undesirable health outcomes.

Bolton-Maggs et al. [66] and Jackson [27] focused on particular cross-sections of the population in order to determine the relationship between attitude and uptake. Bolton-Maggs surveyed English university students about their perceptions of MMR and their current vaccination status, concluding that misconceptions about MMR remained prevalent and those with poor understanding of the diseases were less likely to be vaccinated, particularly if they were male and/or not registered with a GP.

Jackson et al. [27] conducted a qualitative interview study of 174 travellers from GRT communities, and paired this with detailed socio-demographic information on housing status and vaccination data. As with the other focus groups of minority groups, this study highlighted that acceptance of vaccines was generally high and low uptake was self-reported to stem primarily from access issues such as language barriers, illiteracy, lack of housing, or a lack of established, trusting relationships with healthcare providers.

Edelstein et al. [67] aimed to determine whether recent declines in childhood vaccinations since 2012 could be directly linked to strong anti-vaccine attitudes, using a descriptive study which triangulated vaccine coverage data (from the Cover of Vaccination Evaluated Rapidly (COVER) dataset) with a cross-sectional survey of vaccine attitudes (as published in a PHE longitudinal survey) as well as UK-specific Twitter data, used as a proxy measurement for online anti-vaccine attitudes. The authors concluded that no such direct link could be drawn; especially as recent declines in MMR coverage have occurred just as deliberately anti-vaccine attitudes (as measured online and in qualitative surveys) have also declined in the UK. Instead, access and healthcare service issues are more likely to be the cause of recent declines in uptake.

Finally, Brown et al. [68] reported a generic cross-sectional survey using the Brown psychometric tool [63], with a focus on parents reached during a 2008–9 catch-up campaign. The only independent predictors of successful catch-up were having a younger child and perceiving MMR to be socially desirable (no other variables were significant). However, the catch-up campaign was generally successful in inducing parents to catch up on missed vaccines. Taken as a group, these hybrid studies integrate disparate data sources and, as a result, lead to conclusions that broadly support (but occasionally also subvert) our expectations, especially regarding the relationship between education, media consumption, and uptake.

### 4.2. Summary

Firstly, the uptakes and demographics studies found some broad group-level or demographic factors that have strong associations with MMR uptake but crucially different groups had different reasons for lower uptake such as the example by Pearce et al. [32] that unvaccinated minority and low-income groups had more access issues, while unvaccinated affluent families were more likely to have consciously rejected vaccination. Secondly, the beliefs and attitudes studies aimed to uncover individual vaccination motivations and reasons, in comparison to the mainly-quantitively focused and survey or data-based research. These studies focused on specific minority groups predicted to be at higher risk for low uptake, where access issues were particularly highlighted, as well as broader population samples. Unclear information and the Wakefield controversy was shown to drive decisions, as shown with a study by McHale et al. [51]. Thirdly, the healthcare worker focused studies, many of which were based on interviews, highlighted a lack professional confidence in having conversations about vaccination and clarity about their role [52,56]. Fourthly, experimental and psychometric intervention studies presented new tools to support informed decision making about the MMR vaccine, such as web-based or online decision aids [57,58,59]. Studies also found in-school campaigns were effective [61] and demonstrated how improving uptake in an ethnically mixed and socioeconomically deprived community in urban East London [29] could be achieved through a range of strategies (care packages, financial support, higher quality healthcare, demographic research on under-vaccinated groups, and follow-up processes). Finally, mixed methods studies combined quantitative and qualitative data, including questionnaires and measurement instruments, interviews, and twitter data to draw out the relationships—for example Brown et al. [68] found independent predictors of successful catch-up were having a younger child and perceiving MMR to be socially desirable. See Table 1 below for a breakdown of the themes by paper.

## 5. Discussion

### 5.1. Grouping

In grouping these studies, we defined five categories, based on their associated methodology or focus: uptake and associated demographic studies, qualitative studies on beliefs and attitudes, health worker focus studies, experimental and psychometric intervention studies, and combined qualitative and quantitative or mixed-method studies. It is worth commenting on the relative strengths and limitations of these groups.

The quantitatively focused uptake studies benefit from sizable public health datasets and strictly numerical variables. All but two use data from 10,000+ children to determine potential associations with other quantitative and demographic variables such as income level. However, gathering data from a large cohort naturally makes it more challenging to gather detailed individual information, and thus some nuances may be masked; only three of the studies in this group included extensive socio-demographic information outside of the key variables of interest, as a result of either costly data collection efforts [32,38], or the combination of different datasets using geographic information about areas of deprivation and other associated socioeconomic trends [35]. Nonetheless, this subset of purely quantitative methodologies prove useful, especially in elucidating trends with respect to the relationship between socioeconomic variables, education levels, and uptake. Occasionally, they also revise expectations: Hungerford’s [34] conclusion that low income has always been associated with low uptake, even throughout the Wakefield controversy, is a striking result that calls into question common assumptions about vaccine hesitancy. Similarly, Osam’s work describing the impact of maternal mental illness on uptake suggests more focus should be given to this area [38].

The qualitative group can suffer from the opposite problem: the resource-intensive data collection required means that the sample sizes included are generally much smaller, and might also be due to the consideration of data saturisation, which can be judged through an iterative data collection methodology. Although the stated goals of this group of studies are different, with the objective of understanding structural barriers to vaccination [41], documenting key themes in vaccine hesitancy [49], or exploring parental testimony in general [23], it is still notable that less than one-quarter included sample sizes larger than 30. Nonetheless, these studies generally made efforts to gather data about media consumption, education, trust in and knowledge of vaccines, and the surveyed individual’s relationship with HCWs, all of which have been identified as key determinants in vaccine hesitancy more broadly [3]. Qualitative-focused work can also play a role in determining the quantitative questions to be asked or can clarify a quantitative trend that is not understood. The poor uptake trends in GRT communities identified by Maduma-Butshe and McCarthy [69], for example, have defied an exact explanation (although it is likely that such groups have limited engagement with health services because of their mobility and other cultural factors), but thanks to testimony gathered by Newton and Smith [41], Bell et al. [42,43], and Ellis et al. [44], there is better knowledge about this underserved community.

Those that we have categorised as mixed-methods combined the methodologies. Walsh and colleagues sent surveys to groups that already had detailed information in a Child Health database and were particularly affected by the 2012–13 measles outbreak in Wales, thereby facilitating a combination of qualitative data from the mailed surveys with epidemiological data on the outbreak and sociodemographic data from the Child Health dataset [64]. Anderberg et al. combined area-level uptake data from PHE with Health and Social Care Services (HSE) data, which has more extensive socioeconomic and qualitative information [65]. Other mixed-methods studies combine datasets in order to build predictive psychometric tools, such as in Brown et al. [63], which combined Child Health system data with uptake data and a qualitative survey to build a predictive instrument for further research. In general, these studies are complicated to undertake and require access to potentially disparate datasets or complex surveys. However, we note that their mixed-methodology led to more probing conclusions, as well as observations that would not have been possible without the integration of several different sources of information. More researchers should consider this approach in the future.

### 5.2. The Wakefield Controversy

For much of the past decade, research on vaccine attitudes and uptake in the UK has focused on the lingering effects of the Wakefield controversy, especially in light of large measles outbreaks, such as the 2012–13 outbreak in Merseyside [51] or in South Wales [64]. These outbreaks can be directly attributed to a specific cohort with low MMR vaccine uptake following Wakefield’s 1998 paper and the ensuing prolonged media controversy; an outcome that was predicted at the time [70]. In most of the country, first-dosage MMR vaccine uptake at 24 months reached its nadir of ~81% in 2003–4 (directly after the peak of the controversy) before slowly beginning to rise again, but did not reach 90% until 2011 [28]. In essence, while intense and sensationalised media coverage of the Wakefield controversy declined over time, the effects on the public’s perception of the safety of the MMR vaccine persisted and affected an entire cohort of children, especially those born approximately between 1998 and 2004. Catch-up campaigns have had success in alleviating the detrimental effects of this low-uptake period [63], but measles outbreaks will continue to be a threat into the 2020s, particularly because of the combined risk of imported cases from other under-vaccinated countries due to increasing globalisation and migration flow to the UK [28].

As a result, while it is reassuring that MMR vaccine uptake has broadly improved over the last decade [71], research during this time has continued to indicate the ‘long tail’ of the Wakefield controversy for two reasons: first, it had profound long-term effects on public understanding, and is cited throughout the 12 qualitative studies as a common reference point for parents (in many interviews across all groups, Wakefield, autism, and desires for a ‘single’ of ‘safer’ vaccine were mentioned specifically). Second, these negative attitudes have reduced uptake well into the 2010s and are therefore an important factor to consider in any uptake studies, and when planning to mitigate hesitancy around new and future vaccines. This decade’s research makes clear that the shadow of the Wakefield controversy continues to affect MMR vaccine uptake in the UK. However, it is worth noting that most of the studies published from 2017 onwards have focused on a particular demographic group and current research has greater awareness of socioeconomic factors than it has in the past. This is likely because there is increasing evidence that socioeconomic variables remain highly pertinent in explaining current attitude and uptake trends. Conclusions from unstratified sampling of the national population, while useful during a national crisis such as the Wakefield controversy, are now less relevant than those resulting from more demographically-focused research. Additionally, because relatively high national uptake has been restored, qualitative research on under-vaccinated populations now focuses on smaller relevant subsets of the population.

### 5.3. Socioeconomic Variables Mediate Every Stage of Vaccination

The importance of socioeconomic variables is, arguably, the most significant conclusion of this review and is elucidated well by Pearce et al. [32] in their study of a specific 2013 catch-up campaign. They found that parents who were successfully reached by the campaign were predominantly those who had experienced practical barriers in accessing their initial (on-schedule) vaccine and were disproportionately composed of disadvantaged or ethnic minority groups. Contrastingly, those who continue to evade catch-up campaigns disproportionately come from advantaged groups; they were conscious and are continued objectors to MMR vaccination. This reflects a crucial conclusion of this literature writ large: explaining the vaccination status of a child in the UK requires both an understanding of their parents’ attitudes towards MMR vaccination, and also of their socioeconomic characteristics, because these will mediate their access to the vaccine and how their parents think about it, as well as the relative role of each in the final vaccination outcome.

For example, access plays a direct role in uptake (all other variables being held equal), particularly in certain minority communities. Perry et al. [36] found that children of asylum seekers display substantially reduced uptake; their odds of being vaccinated against key infections is around three times lower than the general population, likely because they are not as well engaged by public health services [72]. Osam et al. [38] found that children of mothers with alcohol or substance use disorders are half as likely to be fully vaccinated compared with children of mothers without such disorders, even when adjusting for other confounders. In PHE’s measles and rubella elimination strategy, reaching these minority, at-risk and under-privileged communities is key to the overall effort for elimination; access, and not attitude, is cited as a key barrier to vaccination in the 2020s, especially as general public trust in MMR has largely recovered [28]. A case study included in the 2019 PHE action plan provides information concerning GRT communities, citing literature showing that 63% of measles outbreaks in the Thames Valley region between 2006 and 2009 occurred in these groups (a risk 100-fold higher than the general population) and that GRT children have significantly lower rates of vaccination uptake [69]. Qualitative studies included in this review provide crucial contextual information for this trend: Newton and Smith [40,41] conducted interviews with 16 site-dwelling GRT women, finding that attitudes towards MMR vaccination did not especially differ from the general population, and that their uptake of the vaccine was primarily rooted in the inflexibility of the healthcare system towards their primarily mobile and isolated lifestyles, paired with long-standing issues of discrimination and racism; and this has been supported by further studies [42,43,44].

Other than socio-cultural and domestic context, income and education also play a large role in access and attitudes: recall the broad conclusions of Sandford et al. [39], Haider et al. [37], Hungerford et al. [34], and Baker et al. [35], that socioeconomic deprivation is positively correlated with reduced uptake and timeliness of MMR vaccination. Haider et al. [36] notes that this, remarkably, is a reverse of the pattern that was well documented in the 2000s; in comparable studies conducted at the height of the Wakefield controversy, affluent groups were the least likely to vaccinate and had seen the fastest reductions in uptake [73]. Chang writes about how vaccine denial is maintained once public health information is corrected, presenting evidence that more highly educated mothers responded more strongly to the controversy [74].

Anderberg et al. [65] propose an interesting hypothesis to explain this reversal, concluding that education level (and, in a separate measure but to a lesser degree, household income) stratified how the Wakefield controversy affected parents’ attitudes—uptake by highly-educated and high-income parents declined much more rapidly following the controversy compared to other groups. Anderberg and his colleagues hypothesise that this effect occurs because educated parents absorb health-related information more quickly, to such a degree that a previously-positive association between education/income and MMR status (in the early 1990s) had reversed to an inverse relationship when false information about MMR vaccines was being disseminated. However, one may infer that the same could also be true in the reverse; as the Wakefield controversy subsided and mounting evidence accumulated that MMR vaccines are safe, well-educated and wealthier parents absorbed the new safety information quickly, while those with lower education or language barriers continued to show reduced uptake.

This conclusion would also be consistent with the qualitative studies conducted by Bell et al. [42,43], Tomlinson et al. [45], and Jackson et al. [27], who all found that difficulty in understanding safety information due to language issues was a key factor in the reduced uptake in the UK’s Polish, Romanian, Somali, and GRT communities, especially as even small amounts of doubt can potentially lead to reduced uptake of vaccination [50]. Walsh’s study, which directly related education level and media consumption in Wales with a particular 2012 measles outbreak, provides further support for Anderberg’s hypothesis. In this case study, parents who consumed more English media and had higher educational attainment at the time of the Wakefield controversy were much more likely to avoid MMR vaccination, as a direct result of absorbing negative information about the vaccine (such as the purported link to autism), which then placed their children at heightened risk of contracting measles during the outbreak a decade later [64].

Intriguingly, there is evidence that a similar education and uptake gradient change also occurred in the United States, although notably that disparity continues to persist years after the purported link has been refuted. It is hypothesised that this is because general media coverage of the purported link continued publicly for several years longer than in Europe, and even coverage about the safety of the vaccine paradoxically only reminds viewers of the purported link [74]. Indeed, the recovery of uptake in high-income groups in the UK since the height of the controversy appears to have revealed underlying access inequalities that were always present; Hungerford et al. [34] argue that deprived areas have always seen lower uptake throughout the controversy and beyond, at least based on a dataset of 72,000 children in Liverpool. In summary, the correlation between affluence and vaccination is generally positive, but has seen occasional and unexpected reversals because, when public health information is disseminated, well-educated and high-income parents are more receptive [65].

Over and above this trend, however, this research presents ample evidence that parental socioeconomic characteristics and education levels are important mediating factors at every stage of a child being vaccinated, from beliefs about the vaccine to the ability to access it. The evident importance of these factors for an understanding of how uptake has varied over time indicates that efforts should be made to include education and income level data in future MMR vaccine research, whether qualitative or quantitative. It is worth emphasising that, within this set of 40 studies, many did not consider socioeconomic (education and income) factors at all; a portion measured one or the other, but few measured both variables. Prospective research should rectify this where possible.

### 5.4. Adressing Uptake and Attitudes

In an extensive systematic review of vaccine hesitancy literature published from 2007 to 2012 (>1100 articles), Larson et al. concluded that determinants of vaccine hesitancy, and of an individual’s likelihood of being vaccinated, are extremely complex and context-specific, varying across time, place, and vaccine [75]. The body of literature gathered in this more focused critical review, of MMR-specific publications in the UK of the last decade, further supports this conclusion. Even when gathering literature within one geographical area, on one vaccine, and within one time period, most evidence points to a complexity that is at odds with our general conception of so-called “anti-vaxxers” as a homogenous group to be vanquished with, for example, a one-size-fits-all policy of compulsory vaccination or punitive penalties [4]. Indeed, a study aimed precisely at identifying an association between deliberately anti-vaccine sentiment and uptake decline in the UK was unable to do so [67].

Instead, this review identifies numerous group and individual level determinants, operating directly and indirectly to factor into vaccination outcomes. The consideration of group-level and individual-level determinants goes beyond typical bioethical principles of individual autonomy versus group justice. At the group level, income and especially education are determinants of access, utilisation, and engagement of health services and information, and socioeconomic deprivation plays a strong role to reduce vaccine uptake [34,35,37,39]. These socioeconomic factors also influence the ways information (e.g., the misinformation circulated during the Wakefield controversy) is integrated into health decisions [65]. Secondly, immigration status, and related language barriers but also limited trust of, and access to, British healthcare, introduces barriers to vaccination that antedate the first GP appointment [36,40,42,45,57]. Finally, maternal mental illness has an underexplored impact on uptake not yet fully explained [38]. These access issues are often ignored within the public “anti-vax” media discourse, possibly because they are more hidden, whereas public debates about outright vaccine refusal are more overt [3]. However, it is encouraging that PHE’s elimination strategy places access issues front and centre, particularly because that is likely where the government can have the most success in changing outcomes and have stronger obligations [28,32].

Beyond the issue of access, what has a decade of research told us about the determinants of attitude—assuming one is not affected by prior barriers? A consistent theme of qualitative interviews is misinformation or lack of knowledge and trust in the MMR vaccine, largely stemming from the continued effects of the Wakefield controversy [23,46,48,50]. Vaccines and their risks relative to the diseases they protect against are often misunderstood, especially due to the success of immunisation campaigns: adolescents and university students, for example, have little understanding of MMR, pointing to ineffective education [49; 66]. Following a measles outbreak, parents often report a desire to be more informed about the dangers of such diseases [51]. Consumption of negative media, particularly in times of high controversy can also reduce uptake [64].

However, governmental information is often mistrusted, and legislative interventions elicit negative responses in focus groups [47]. Even HCWs, expected to encourage vaccination, may perceive vaccines as less important than they really are [56] and often report that they do not have adequate tools or skills to communicate appropriate information to parents [52]. This is unfortunate, as it has been suggested HCWs (particularly practice nurses and health visitors), are seen as credible sources of information by parents [21,54]. We are not suggesting a wholesale change in the role of HCWs—particularly nurses, school nurses, and health visitors in encouraging vaccination. However, it is consistently reported that they do not feel well-equipped to deliver vaccination education as a part of their role.

All of these factors contribute to a child’s vaccination status, and this complexity may make it appear that government campaigns waged to raise uptake (such as those mentioned in the 2019 PHE report) are quixotic—after all, it is unlikely that a vaccine campaign can erase long-standing social inequalities. However, research presents promising new advances to be integrated into coherent public health strategy. Predictive psychometric tools have been developed which can determine who is likely to avoid vaccinations based on simple questionnaires, allowing for more directed campaigns [62,63]. Interventional tools have also had success in changing vaccination outcomes in preliminary trials, including online based decision-aids [57,59,60]—these have subsequently been confirmed as cost-effective and could see widespread distribution [76]. Finally, catch-up campaigns are successful, particularly in reaching groups with limited access to health-care and whose lack of MMR vaccination was not necessarily a conscious decision [68]; these may be implemented through text message reminders [77] or the help of school nurses [61]. One trial, which gave monetary incentives to general practices that exhibited success, used robust information technology and follow-up processes, and invested in detailed demographic study of a particularly deprived area of London, increased uptake of MMR vaccination by almost 15% in two years [29]. In summary, this body of research makes clear that, while successful MMR vaccine uptake is a highly complex issue, it is is an achievable goal.

## 6. Conclusions

We began this review by providing a background to MMR vaccination, outlining policy changes, trends in uptake, as well as cases and outbreaks in order to provide the context for current understanding. We then organised the results of the review into themes to highlight the uses of different methodologies, including the advantages and disadvantages in the methodology utilised. Uptake and demographic studies were insightful for understanding group-level associations; beliefs and attitudes studies for vaccination reasons and motivations; HCW studies about barriers and improvements for promoting vaccination; experimental and psychometric intervention studies for assessing informational tools and strategies; and mixed methods studies for drawing out the relationship between attitudes, access, and uptake. A key event that was referenced across the studies was the Wakefield controversy, which we revisited in light of the studies from this review. We also discussed the importance of socioeconomic variables, in how access plays a direct role in uptake in certain minority communities [36], while attitudes of advantaged groups are driving factors for evasion of catch-up campaigns [32].

Although there will be differences according to vaccination and timing, there is value in taking the lessons from this review to future immunisation campaigns. It is anticipated that the next stage of the for COVID-19 immunisation programme will move to vaccinating children, and the lessons from MMR vaccination will become pertinent. While deployment of COVID-19 vaccines to at-risk HCWs and vulnerable older age groups in the UK has been proclaimed a success, the challenges of vaccinating children, who are at lower risk of serious forms of the disease, is yet to be seen. It is likely in such an evolving scenario that quantitative studies on demographics and uptake will be relied on, supplemented by qualitative work surveying parents. Communication should use trusted HCWs and address controversy early on to avoid ongoing misinformation or a lack of information and trust that was seen from the Wakefield controversy. Learning from MMR research, the need to consider both access and attitudes is therefore evident, as is the need for a greater focus on access issues pertaining to various groups in society according to education, income, and immigration status.

## Figures and Tables

**Figure 1 vaccines-09-00402-f001:**
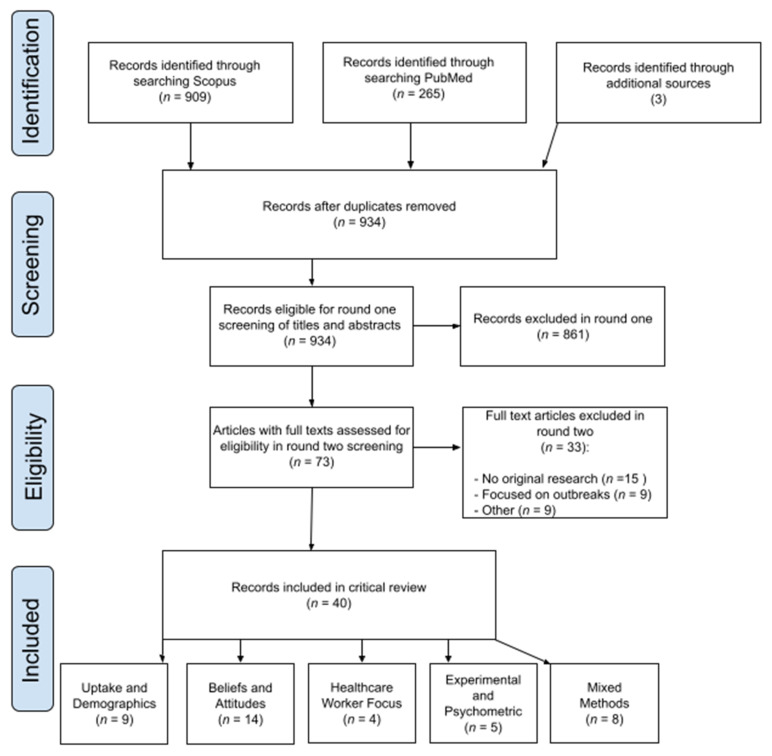
Adapted from [22] Moher D, Liberati A, Tetzlaff J, Altman DG, The PRISMA Group (2009). Preferred Reporting Items for Systematic Reviews and Meta-Analyses: The PRISMA Statement. PLoS Med 6(7): e1000097. doi:10.1371/journal.pmed1000097.

**Table 1 vaccines-09-00402-t001:** List of papers by theme.

Study Group	Author	Year	Summary
Uptake and Demographics	Baker	2011	Nine papers were largely quantitative, consisting of surveys or analyses focusing only on uptake of the MMR vaccine, and potential inequalities or predictive demographic factors therein
	Emerson	2019	
	Haider	2019	
	Hungerford	2016	
	Hutchings	2016	
	Osam	2020	
	Pearce	2013	
	Perry	2020	
	Sandford	2015	
Beliefs and Attitudes	Bell	2019	The second and largest group of eligible literature comprises 12 studies relating to beliefs and attitudes, typically more qualitatively-focused, using smaller sample sizes and person-to-person interviews or qualitative surveys to contextualise the path to vaccination on an individual level
	Bell	2020	
	Brown	2012	
	Ellis	2020	
	Gardner	2010	
	Hill	2013	
	Hilton	2013	
	Johnson	2014	
	Kennedy	2014	
	McHale	2016	
	Newton	2017	
	Smith	2017	
	Tickner	2010	
	Tomlinson	2013	
Healthcare Worker Focus	Bell	2020	Four papers focused exclusively on the attitudes of HCWs themselves towards MMR vaccination
	Hill	2021	
	Mytton	2013	
	Redsell	2010	
Experimental and Psychometric	Altinoluk-Davis	2020	Five papers were experimental in nature, with three pertaining specifically to new tools developed to support informed decision-making about the MMR vaccine
	Cockman	2011	
	Jackson	2010	
	Jackson	2011	
	Shourie	2013	
Mixed Methods	Anderberg	2011	Eight papers were mixed methods in their approach, comprising a combination of quantitative uptake data with a qualitative understanding of individual knowledge, sociodemographic variables, or decision making.
	Bolton-Maggs	2012	
	Brown	2011	
	Brown	2011	
	Edelstein	2020	
	Jackson	2017	
	Tickner	2010	
	Walsh	2015	

## Data Availability

No new data were created or analyzed in this study. Data sharing is not applicable to this article.

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
