# Peer review of "MMR Vaccine Attitude and Uptake Research in the United Kingdom: A Critical Review"

_vaccines, 2021, doi:10.3390/vaccines9040402_

Round 1

Reviewer 1 Report

An interesting critical review study on the attitudes/uptake of the MMR vaccine in the UK.

Title: Relevant and appropriate.

Introduction: Thorough, well-written, but the research questions (l. 47-48) should go at the end of the section. According to the authors, “We will focus on MMR uptake and attitude research in the United Kingdom …”, however, the inclusion criteria (p. 4, l. 159-160) include attitudes, decision-making, uptake or beliefs; thus these terms should be unified for easier understanding.

The strategy of ‘citation-chasing’ is a strong point in order to not miss any studies. Methods: Researchers extracted the data using a grounded theory approach (p.4, l. 182), this criteria should be explained. Why was it done that way? What are the steps? How does that influence emerging themes?

In Section 3.2, were there any disagreements between the researchers who extracted the data? How were these resolved? While the authors explain their use of CASP, they should explain the quality criteria used for choosing publications.

Objective: Although it can be understood in the wording of the article, there should be a specific section at the end of the background which states, "The objective of this study is ..."

Results: Interesting, comprehensive and well-described. The authors describe the studies in five sections, which is a bit excessive, but while this may be acceptable, they should also include an inductive study with themes emerging from the analysis. The information about each article is quite long and should be summarised. Health visitors are a local figure, unknown outside of the UK, thus, their functions should be described and the difference between this person’s role and doctors and nurses’ role (p. 8, l. 325-330).  This is also the case with healthcare workers and their functions regarding vaccinations.

Discussion: Comprehensive and appropriate, the authors opt for mixed methods, qualitative studies with larger sample sizes, on under-vaccinated populations, socioeconomic characteristics, issues with access to minority communities and parents’ attitudes as key elements to children’s vaccination status. How do you explain aspects such as vaccine denial among the upper class/well-informed? (p. 13, l. 585-598). The relationship between the onslaught of information and fake news and a lower acceptance rate of vaccination is one aspect that should be explained further. Discuss the repercussions of your review with the bioethical principles of justice-autonomy. Nurses are responsible for health education, which is relevant for public health strategy. Are the authors suggesting that the role of nurses, school nurses, and visitors in providing vaccination information should be reviewed? Methodological limitations should go to the end of the discussion section.

Conclusions: Quite long, and may be repetitive. Write a section for clear, succinct conclusions, which answer your questions. Using bullet points may be useful. Your study should end with clear and concrete implications for practice, for example, what have we learned in order to apply this information to the COVID-19 vaccination campaign? 

Author Response

Review 1

Response

An interesting critical review study on the attitudes/uptake of the MMR vaccine in the UK.

Thank you, we appreciate this feedback

Title: Relevant and appropriate.

Introduction: Thorough, well-written, but the research questions (l. 47-48) should go at the end of the section. According to the authors, “We will focus on MMR uptake and attitude research in the United Kingdom …”, however, the inclusion criteria (p. 4, l. 159-160) include attitudes, decision-making, uptake or beliefs; thus these terms should be unified for easier understanding.

We have moved the questions to the end of the introduction and better unified the inclusion criteria:

“Search queries were made using PubMed and Scopus, with terms aimed at capturing all relevant literature focusing on attitudes and uptake, as well as closely connected subjects of access, decision-making, or beliefs about the MMR vaccine in the UK”

The strategy of ‘citation-chasing’ is a strong point in order to not miss any studies. Methods: Researchers extracted the data using a grounded theory approach (p.4, l. 182), this criteria should be explained. Why was it done that way? What are the steps? How does that influence emerging themes?

We have better explained the grounded theory approach: “Two researchers analysed the data using a grounded theory approach, identifying themes as they emerged to produce an explanatory framework. This was an iterative process, to generate themes that emerged from the data. Early on we recognised a main organising factor in the literature was between the methodological approach taken. We first started with the largest group of quantitatively-focus uptake studies and then followed the grouping from there. We decided upon this approach as similar literature reviews of MMR aimed to answer questions that resulted in pre-determined themes. We were more concerned with allowing the data to determine the results, following on from our starting point of two areas of focus ‘uptake and attitudes’ which already involved a different methodological treatment.”

In Section 3.2, were there any disagreements between the researchers who extracted the data? How were these resolved? While the authors explain their use of CASP, they should explain the quality criteria used for choosing publications.

We have added the following: “The data extraction was checked by another author, with no masking used. Any disagreements were resolved through discussion through regular meetings during the analysis stage.”

Objective: Although it can be understood in the wording of the article, there should be a specific section at the end of the background which states, "The objective of this study is ..."

We have included a section stating the objective: “Our objective is to produce a timely review of what we know about Britain’s experience of MMR, and a multi-disciplinary understanding of the factors influencing an individual’s vaccination status.”

Results: Interesting, comprehensive and well-described. The authors describe the studies in five sections, which is a bit excessive, but while this may be acceptable, they should also include an inductive study with themes emerging from the analysis. The information about each article is quite long and should be summarised.

We have provided a short summary. See 4.2 Summary.

Health visitors are a local figure, unknown outside of the UK, thus, their functions should be described and the difference between this person’s role and doctors and nurses’ role (p. 8, l. 325-330).  This is also the case with healthcare workers and their functions regarding vaccinations.

We have included a description: “Health visitors are a specialised healthcare role directed at community health in the UK and often visit parents and their children at home. They are nurses or midwives who concentrate on the health of pre-school age children, including ensuring they are vaccinated.”

Discussion: Comprehensive and appropriate, the authors opt for mixed methods, qualitative studies with larger sample sizes, on under-vaccinated populations, socioeconomic characteristics, issues with access to minority communities and parents’ attitudes as key elements to children’s vaccination status.

Thank you, these were the key aspects we identified for the discussion.

How do you explain aspects such as vaccine denial among the upper class/well-informed? (p. 13, l. 585-598).

To explain aspects such as vaccine denial among the upper class/well-informed – we make clearer our discussion on how vaccine denial can arise (particularly by laying out in more detail the Anderberg hypothesis that the highly educated retain health information more quickly):

“These results go some way to explain aspects of vaccine denial among those who are more highly educated. Strikingly, the study found that uptake declined faster in areas of higher education, with spill-over effects to other vaccines, contrary to the more familiar pattern in which low educational achievement and socioeconomic deprivation correlate with undesirable health outcomes.”

The relationship between the onslaught of information and fake news and a lower acceptance rate of vaccination is one aspect that should be explained further.

We further explain the information and fake news that leads to a lower acceptance rate by better bringing out our discussion of the Chang paper:

“Chang writes of about how vaccine denial is maintained once public health information is corrected, in presenting evidence that more highly educated mothers responded more strongly to the controversy [72].”  

Discuss the repercussions of your review with the bioethical principles of justice-autonomy.

We have included a note in the discussion: “Instead, this review identifies numerous group and individual level determinants, operating directly and indirectly to factor into vaccination outcomes. The consideration of group-level and individual-level determinants goes beyond typical bioethical principles of individual autonomy vs group justice. At group levels, income and especially education are determinants of access, utilisation, and engagement of health services and information, and socioeconomic deprivation plays a strong role to reduce vaccine uptake [33; 34; 36; 38]. These socioeconomic factors also influence the ways information (e.g. the misinformation circulated during the Wakefield controversy) is integrated into health decisions [64].”

Nurses are responsible for health education, which is relevant for public health strategy. Are the authors suggesting that the role of nurses, school nurses, and visitors in providing vaccination information should be reviewed?

We outline our view of the role of HCWs in from our review: “We are not suggesting a wholesale change in the role of HCW ­– particularly nurses, school nurses, and health visitors in encouraging vaccination. However, it is consistently reported that they do not feel well-equipped it is to deliver the vaccination education that is already a part of their role.”

Methodological limitations should go to the end of the discussion section.

We have moved the methodological limitations to the discussion section.

Conclusions: Quite long, and may be repetitive. Write a section for clear, succinct conclusions, which answer your questions. Using bullet points may be useful. Your study should end with clear and concrete implications for practice, for example, what have we learned in order to apply this information to the COVID-19 vaccination campaign? 

We have edited the conclusion to remove the repetition and make more succinct. We have added implications for practice:

“Communication should use trusted HCWs and address controversy early on to avoid ongoing misinformation and a lack of trust that was seen from the Wakefield controversy.”

Reviewer 2 Report

Great paper, honestly. Very well written, well researched, excellent conclusions. I can usually come up with something to harp on but it's great

111: Might be nice to include something about Wakefield’s medical license being revoked or original papers being withdrawn

133: maybe a date range

208: maybe a ref for CASP

222: I’m going to suggest designing a new figure to address the results. A table that condenses down each of your 5 major themes could be nice, so one big table, each theme as a big heading, then columns addressing each paper. Some ideas: number of participants, demographics, time period, location, ref #

504: Maybe consider a different title than wakefield looms large

598: your conclusions supporting this prior work is spot-on

Your conclusions are well taken, particularly the heterogeneity of anti-vaxxer attitude determinants. Overall this is a superb article

Author Response

Review 2

Great paper, honestly. Very well written, well researched, excellent conclusions. I can usually come up with something to harp on but it's great

Thank you – these comments are really appreciated!

111: Might be nice to include something about Wakefield’s medical license being revoked or original papers being withdrawn

We have noted this: “The 1998 the paper by Wakefield and colleagues in The Lancet was retracted but the damage had already been done.”

133: maybe a date range

We have added ‘from 2010 to 2021’.

208: maybe a ref for CASP

We have added a reference.

222: I’m going to suggest designing a new figure to address the results. A table that condenses down each of your 5 major themes could be nice, so one big table, each theme as a big heading, then columns addressing each paper. Some ideas: number of participants, demographics, time period, location, ref #

Thank you for suggesting, please see the new results:  4.2.1 Table 1

504: Maybe consider a different title than wakefield looms large

Good suggestion – we have changed this to the ‘The Wakefield controversy’.

598: your conclusions supporting this prior work is spot-on

Your conclusions are well taken, particularly the heterogeneity of anti-vaxxer attitude determinants. Overall this is a superb article

Thank you so much.

Reviewer 3 Report

Estimated Authors,

Estimated Editors,

I've read with great interest the paper "MMR Vaccine Attitude and Uptake Research in the United Kingdom: A Critical Review".

This article focuses on the various factors involved in vaccine acceptance: available evidence is carefully summarised, and properly discussed. In facts, despite the substantial complexity of this topic, Torracinto et al. lead their readers across a lot of data and disparate experiences, including a throughtful discussion of the "WAKEFIELD" case.

In summary, I could endorse the final acceptance of this article: my only request is to clarify in the methods section why Authors have opted for the timeframe 2010-2021, in particular: why starting with 2010? It is particularly important as the Authors (in the limitation subsection) specifically include the cut-off date of 2010 as a sort of limitation as a substantial evidence base was reported shortly before.

Author Response

Review 3

I've read with great interest the paper "MMR Vaccine Attitude and Uptake Research in the United Kingdom: A Critical Review".

This article focuses on the various factors involved in vaccine acceptance: available evidence is carefully summarised, and properly discussed. In facts, despite the substantial complexity of this topic, Torracinto et al. lead their readers across a lot of data and disparate experiences, including a throughtful discussion of the "WAKEFIELD" case.

Thank you for your encouraging feedback.

In summary, I could endorse the final acceptance of this article: my only request is to clarify in the methods section why Authors have opted for the timeframe 2010-2021, in particular: why starting with 2010? It is particularly important as the Authors (in the limitation subsection) specifically include the cut-off date of 2010 as a sort of limitation as a substantial evidence base was reported shortly before.

We have addressed this important point in the methods:

“We reviewed data over a ten year period starting from 2010. The reason for choosing this start date was because by 2010 MMR uptake was beginning to recover after an all-time low in 2003 but because of the time lag between uptake and cases, early 2012 still saw the largest outbreak of measles since 1988 (which also resulted in an increase in MMR uptake rates). Therefore, this time period avoids the earlier literature focusing more directly on uncertainty about the Wakefield claims in the 1990s and subsequent fall-out in the early 2000s, although much the literature we assessed still referenced these earlier events.”

We have also clarified that although we identified the time frame as a limitation, on balance this was to produce a more focuses review:

“A second limitation is the timeframe and geographical constraints of the search query; there is an abundance of high-quality research that was published prior to 2010 or was not conducted in the UK but remains relevant to our current understanding of MMR attitudes and uptake. While not included in this review, this earlier work still provides a foundation for this decade’s research, and so still influences this synthesis. Therefore, we decided that the literature after 2010 was better focused on questions of uptake and attitudes, by that time there had been a refutation of the Wakefield claims and MMR coverage had begun to recover.”